# New Hybrid Nanofiltration Membranes with Enhanced Flux and Separation Performances Based on Polyphenylene Ether-Ether-Sulfone/Polyacrylonitrile/SBA-15

**DOI:** 10.3390/membranes12070689

**Published:** 2022-07-04

**Authors:** Gabriela Paun, Elena Neagu, Viorica Parvulescu, Mihai Anastasescu, Simona Petrescu, Camelia Albu, Gheorghe Nechifor, Gabriel Lucian Radu

**Affiliations:** 1National Institute for Research-Development of Biological Sciences, 060031 Bucharest, Romania; gpaunroman@gmail.com (G.P.); lucineagu2006@yahoo.com (E.N.); camelia_barsan2000@yahoo.com (C.A.); 2“Ilie Murgulescu” Institute of Physical Chemistry of the Romanian Academy, Splaiul Independentei 202, 060021 Bucharest, Romania; vpirvulescu@icf.ro (V.P.); manastasescu_ro@yahoo.com (M.A.); simon_pet@yahoo.com (S.P.); 3Faculty of Chemical Engineering and Biotechnologies, University Politehnica from Bucharest, 313 Splaiul Independentei, 060042 Bucharest, Romania; doru.nechifor@yahoo.com

**Keywords:** hybrid nanofiltration membrane, PPEES, PAN, SBA-15, antifouling, concentration performance, polyphenols, flavones

## Abstract

This study presents the preparation of hybrid nanofiltration membranes based on poly(1,4-phenylene ether ether sulfone), polyacrylonitrile, poly(vinyl pyrrolidone), and SBA-15 mesoporous silica. Laser treatment of polymeric solutions to enhance the hydrophilicity and performance of membranes was investigated. The membranes’ structure was characterized using scanning electron (SEM) and atomic force (AFM) microscopy and contact angle measurements. The addition of PAN in the casting solution produced significant changes in the membrane structure, from finger-like porous structures to sponge-like porous structures. Increased PAN concentration in the membrane composition enhanced the hydrophilicity of the membrane surface, which also accounted for the improvement in the antifouling capabilities. The permeation of apple pomace extract and the content of polyphenols and flavonoids were used to evaluate the efficacy of the hybrid membranes created. The results showed that the hybrid nanofiltration membranes based on PPEES/PAN/PVP/SBA-15: 15/5/1/1 and 17/3/1/1 exposed to laser for 5 min present a higher rejection coefficient to total polyphenols (78.6 ± 0.7% and 97.8 ± 0.9%, respectively) and flavonoids (28.7 ± 0.2% and 50.3 ± 0.4%, respectively) and are substantially better than a commercial membrane with MWCO 1000 Da or PPEES-PVP-based membrane.

## 1. Introduction

Membrane technology, particularly nanofiltration, is of great interest in the recovery of bioactive compounds from the agri-food industry, due to mild processing conditions, simple operation, and low energy consumption [1]. Some of the most valuable compounds from agri-food are polyphenols which have antioxidant and anti-inflammatory properties [2]. They have some essential beneficial health effects, such as the prevention of cancer, cardiovascular disease, diabetes, neurodegenerative diseases, etc. [1,2,3]. The concentration of bioactive compounds, such as polyphenols from agro-industrial by-products, is significant for obtaining high-value-added products. Recent studies have shown the effectiveness of nanofiltration in the concentration of polyphenolic compounds from plant sources (herbs, fruits, vegetables) [4,5,6].

Although nanofiltration has been successfully used to concentrate bioactive compounds, it remains a continuous challenge to obtain membranes with competitive separation properties, high permeability, and a better antifouling capacity.

Most of the polymeric compounds utilized in the fabrication of membranes are hydrophobic, causing fouling issues, a permanent decrease in permeate flow, and a shorter lifetime for the membrane [7,8].

The literature on ultrafiltration and nanofiltration membranes strongly suggests that the most widely-used polymers are polysulfone (PS), polyethersulfone (PES), polyacrylonitrile (PAN), polypropylene (PP), polytetrafluoroethylene (PTFE), polyamide (PA), and cellulose acetate (CA) [9,10,11,12,13].

In recent years, poly(1,4-phenylene ether ether sulfone) (PPEES) was studied for its practical application as a separation membrane, mainly due to its high thermal resistance, film-forming capacity, good chemical stability, good solubility, and easy preparation with low-cost [14,15]. Our previous research revealed that PPEES could be used to prepare ultra- and nano-filtration membranes for polyphenolic compound concentration from natural extracts [5,16]. However, due to its hydrophobic nature, PPEES is easily fouled, resulting in a gradual decrease in permeate flux.

On the other hand, PAN membranes are solvent-resistant and have a low fouling character due to their hydrophilic nature [17]. The disadvantages are the poor solubility of PAN in various solvents and low chemical stability [18].

Previous research showed that hydrophilicity and rugosity are the main parameters conditioning the membrane fouling, influencing permeation and separation performances. Therefore, recently, hydrophilic membranes have attracted special consideration in practical use, due to their superior fouling resistance and high permeate flux [18,19].

Current research focuses on developing new types of membranes, mainly to improve flow and reduce fouling. Some methods have been applied to obtain high-performance polymeric nanofiltration (NF) membranes by chemical and physical modification of membrane surfaces with hydrophilic inorganic or organic molecules. One of these methods is the introduction of inorganic compounds in a polymeric solution to form polymeric nanocomposite membranes. It was reported that using inorganic fillers such as (TiO_2_ [20,21], Fe_3_O_4_ [22], ZrO_2_ [23], SiO_2_ [24,25], etc.) in polymeric membranes brings significant changes to the performance of the modified membranes. Among the inorganic fillers, silica (SiO_2_) has been intensively studied to increase the membrane permeances for gases, such as CO_2_ and O_2_ [24,25,26,27] and to enhance the hydrophilicity and antifouling properties of membranes [28]. In particular, SBA-15 (Santa Barbara Amorphous No. 15), a very studied mesoporous silica due to its chemical, thermal, and mechanical stability, could considerably enhance the hydrophilicity of composite membranes at low dope concentration, and it has a low price [29].

The hydrophilicity of membranes can also be increased by the addition of hydrophilic polymers, such as poly (vinyl pyrrolidone) (PVP) or poly (ethylene glycol) (PEG) [30,31].

The effects of laser treatment of polymers on structural changes (morphology, contact angle), surface roughness, and hydrophilicity have been recently published [32,33]. This can be applied as an alternative method for improving the hydrophilicity of the membranes.

In this study, we firstly present nanofiltration membranes with improved hydrophilicity and biological compound retention based on PPEES, PAN, and PVP polymers with SBA-15 mesoporous silica as a nanofiller. The influence of the laser on the polymer solution was also studied to obtain nanostructured membranes with improved performance. Moreover, there is no information on the laser processing of polymeric solutions for membranes. The morphological structure and separation performance of the prepared PPEES/PAN/PVP/SBA-15 hybrid membranes were examined in detail. The membrane’s structure was characterized by scanning electron (SEM) and atomic force (AFM) microscopy and contact angle measurements. Bioactive polyphenols are some of the valuable compounds in apple by-products that have been considered in recent years. For this purpose, the membrane’s permeation and selectivity were assessed via the polyphenol content from this extract.

## 2. Materials and Methods

### 2.1. Materials

Poly(1,4-phenylene ether-ether-sulfone) (PPEES), polyacrylonitrile (PAN), SBA-15 mesoporous silica, Folin-Ciocalteu reagent, sodium carbonate, and sodium acetate were purchased from Sigma-Aldrich Chemical Company, Inc. (Milwaukee, WI, USA). Polyvinyl pyrrolidone K90 (PVP), polyethylene glycols (PEG), and bovine serum albumin (BSA) were provided by Fluka (Buchs, Switzerland). Methanol, 1-Methyl-2-pyrrolidone (NMP), and aluminum chloride anhydrous were acquired from Honeywell (Charlotte, NK, SUA). The reference compounds (gallic acid, chlorogenic acid, ellagic acid, rutin, luteolin, quercetin, quercetin 3-β-D-glucoside, quercitrin, kaempferol, pholoridzin dihydrate, and catechin) were obtained from Sigma Aldrich (St. Louis, MI, USA) and Roth (Carl Roth GmbH, Karlsruhe, Germany). Commercial NF flat sheet membrane made from regenerated cellulose with a molecular weight cut-off (MWCO) of 1000 Da was purchased from Millipore Corporation (Burlington, MA, USA) and used for comparison.

### 2.2. Hybrid Membranes Preparation

The hybrid nanofiltration membranes were prepared using the phase inversion technique. Membranes based on PPEES were obtained using a mixed polymeric-inorganic solution of PPEES, SBA-15 silica, and PAN at a different weight ratio of PPEES:PAN, in NMP solvent, as presented in Table 1. The solution was stirred for 6–8 h at 60 °C until a homogenous solution was obtained. Following, three samples of the mixed polymeric-inorganic solution with composition M1 were subjected to laser treatment under stirring for 2, 5, or 10 min to produce M1-L2, M1-L5, and M1-L10 membranes, respectively, and were subsequently characterized. An infrared laser PSU-H-LED (Changchun New Industries Optoelectronics Tech. Co., Changchun, China) was used at 1550 nm and 200 mW power. Subsequently, based on the results obtained for the M1 membranes, the M2-L5 and M3-L5 membranes were also prepared by laser treatment.

The doped solution was sonicated for 30 min to deaerate it. It was then poured onto a flat glass surface and stretched with a stainless-steel roller to obtain a thin film. The polymer was precipitated in a bath with deionized water where it was kept for about 1 h, then the membrane was dried at room temperature.

### 2.3. Membrane Characterization

#### 2.3.1. Morphology

SEM micrographics performed on samples were recorded using a high-resolution microscope, FEI Quanta 3D FEG instrument (FEI), operating in high vacuum mode with an accelerating voltage of 1 kV. Samples were placed on double-sided conductive tape and scanned without special preparation. The surface roughness of these membranes was investigated in contact mode by atomic force microscopy (AFM) using the XE-100 microscope from Park Systems, equipped with flexure-guided, cross-talk eliminated scanners. All AFM images were recorded with pre-mounted NSC36/Hard Al BS tips (MikroMash) with rotated tip shape, full cone angle of 40°, a radius of less than 20 nm, height ~ 15 mm, 90 mm mean length, 32.5 mm mean width, ~2 N/m force constant, and thickness ~ 1 mm. The recorded AFM images were processed using the XEI program (v 1.8.0-Park Systems) for displaying purpose and roughness evaluation. In order to increase the contrast of the morphological features, the images are presented in the so-called “enhanced contrast” view mode which uses the change of a pixel relative to its neighbors. Representative line scans are presented below the AFM images, showing, in detail, the surface profile of the scanned samples and the dimensions of the selected features (particles, pits, etc.).

#### 2.3.2. Surface Wetting Properties

The hydrophilicity of the membrane’s surface was analyzed through water contact angle measurement using the Young-Laplace (Sessile Drop Fitting) method and a drop shape analysis system, DSA 2 Easy Drop instrument (Krüss GmbH, Hamburg, Germany). The contact angle values of each sample were measured at five diverse positions and the average value was considered the representative result.

#### 2.3.3. Membrane Molecular Weight Cut-Off

Molecular weight cut-off (MWCO) is one of the most valuable tools for characterizing NF membranes. The MWCO of NF membranes is usually in the range of about 200–1500 Dalton. We determined the MWCO for membrane using polyethylene glycols (PEG) with a molecular weight of 400–1500 Da at a concentration of 50 mg/L. The concentrations of PEG were measured by means of BaCl_2_ colorimetry [34].

### 2.4. Evaluation of Membrane Performance

#### 2.4.1. Preparation of Feed Solution

Apple pomace extract as a feed solution was prepared using ultrasonic-assisted extraction (UAE) in a sonication bath (Elma Transsonic T 460, Singen, Germany), at a frequency of 35 kHz for 60 min. The extraction was performed in a 50% (*v*/*v*) ethanol solution, and the concentration of the fresh apple pomace was 30% (*w*/*v*).

#### 2.4.2. Membrane Antifouling Ability

For the evaluation of the antifouling performance of each membrane, bovine serum albumin (BSA) was used as the foulant solution. Membranes were tested using the Cell CF-1 cross-flow lab-scale filtration module at a 5 bar pressure. Initially, distilled water was passed through the membrane for 30 min and the water flow was recorded (J0). Then, a 0.3 g/L BSA solution was used as a feed solution and the flow (Jp) was recorded. After BSA solution filtration, the membrane was rinsed with distilled water for 10–15 min and pure water flow was measured again for 30 min. Flux recovery rate (FRR), total fouling rate (Rt), reversible fouling rate (Rr), and irreversible fouling rate (Rir) could be used to appraise the antifouling efficiency of the prepared membrane using Equations (1)–(4) [35]:(1)$$\mathrm{FRR}=\frac{J1}{J0}\times 100$$
(2)$$\mathrm{Rt}=\left(1-\frac{\mathrm{Jp}}{J0}\right)\times 100$$
(3)$$\mathrm{Rr}=\frac{J1-\mathrm{Jp}}{J0}\times 100$$
(4)$$\mathrm{Rir}=\frac{J0-J1}{J1}\times 100$$

#### 2.4.3. Permeability and Selectivity of Membranes

Performances of the prepared membranes were evaluated by a KMS Laboratory Cell CF-1 cross-flow lab-scale filtration module (effective membrane area of 28 cm^2^). The obtained results for nanofiltration membranes were compared to those of a commercial membrane with a 1000 Da molecular weight cut-off.

The pure water, permeate flux (J), rejection rate (R_j_), and solute permeability coefficient (B) were determined using Equations:(5)$$J=\frac{V}{A\cdot t}\;(L\;m^{-2}\;h^{-1})$$
where *V* is filtrate volume (L), *A* is the effective membrane area (m^2^), and *t* is the filtration time (h). The volumetric concentration ratio (feed volume/retentate volume) was 2.5.

The solute rejection (R, %) was calculated using the following formula: (6)$$\mathrm{Rj}=\left(1-\frac{Cp}{Cf}\right)\times 100$$
where *c_p_* and *c_f_* are the polyphenols concentration in permeate and in the feed solution, respectively.
(7)$$B=\frac{J_{s}}{\Delta C\cdot \exp\left(\frac{j_{w}}{k_{s}}\right)}$$
where J_s_ is solute flux, j_w_ is water flux, ΔC = c*_f_*− *c_p_*, and solute mass transfer coefficient k_s_ was estimated as 100 L m^−2^ h^−1^.

The performance of NF membranes can also be evaluated by measuring the saline rejection at 8 bar. Magnesium sulfate (MgSO_4_) rejection was analyzed using a magnesium sulfate solution with a concentration of 100 mg/L. The magnesium concentration in the feed solution and permeate were determined using the multiparameter photometer HI 83,399 (Hanna Instrument, Woonsocket, RI, USA)).

### 2.5. Analytical Methods

#### 2.5.1. Total Phenols, Flavones, and Antioxidant Potential Analyses

The phenolic content was spectrophotometrically determined using a Folin–Ciocalteu assay [36] and expressed as ellagic acid equivalents (EAE) mg/mL).

The flavonoid content was calculated using the AlCl_3_ colorimetric method as described by Lin [37] and was expressed as quercetin equivalents (QE) μg/mL.

#### 2.5.2. HPLC Determination of Polyphenols

Polyphenols were analyzed using a Shimadzu HLPC system with an SPD-M20A diode array detector. Phenolic acids, flavonols, and flavan-3-ols were identified using a method previously described by Alecu et al. [38]. A Nucleosil 100–10 C18 Kromasil (4.6 × 250 mm) column was used and eluted at 20 °C for 60 min. Water and acetonitrile, adjusted to pH 3.0 with formic acid, were used as the mobile phase. Binary gradient elution was used to separate the phenolic compounds, and quantification was performed based on the absorption maximum of the compounds in the UV-Vis spectrum.

## 3. Results

### 3.1. Characterizations of the Nanofiltration Membranes

The cross-section SEM of M1 nanofiltration membranes showed (Figure 1) variation of the porous structure compared to PPEES membrane (M0) which has a typical asymmetric structure and finger-like pores below the dense top surface, as reported in our previous work [15].

The addition of the PAN and SBA-15 in the casting solution produced significant changes in the membrane structure, from finger-like porous structure to sponge-like porous structure. The increasing of porosity by the appearance of gaps in the membrane structure was evidenced.

It can also be seen that after 2 min of laser treatment there were no major structural changes in the morphology of the cross-section. A longer time (10 min) led to the appearance (M1-L10 sample) of large cavities (macro voids), which is consistent with previous studies [33]. It has been observed that the optimal laser treatment time is 5 min. Therefore, all other hybrid membranes were obtained by laser treatment for 5 min (M2-L5 and M3-L5), following, instead, the effect of the PAN concentration on the hydrophilicity, and implicitly, the membrane performance. Previous studies have highlighted that laser treatment of membranes induces morphological changes and increases their hydrophilicity [39,40], but the influence of the laser on the polymer solution has been little studied [33].

In the case of the typical asymmetric membrane structure, a higher porosity was observed for M3-L5 samples with a higher percent of PPEES and lower percent of PAN. SEM images of cross-sections also show SBA-15, with spherical symmetry uniformly distributed into the polymeric matrix.

In order to examine the topographical characteristics, AFM images have been scanned at (2 × 2) µm^2^ scales. AFM was used to examine, in detail, the peculiar surface morphological features of each sample. Thus, Figure 2a–f present enhanced-contrast 2D AFM images at the scale of (2 × 2) µm^2^ of the investigated samples together with characteristic line scans recorded from each image.

Sample M0 (Figure 2a) exhibited randomly distributed pores, relatively large in diameters, such as the pore along the horizontal green line (with a diameter of ~188 nm). The surface was slightly corrugated with some ridges/crests also visible in the surface profile (line scan) depicted below the AFM image. The root mean square (RMS) roughness of the whole image was 10.6 nm, while the global peak-to-valley parameter (R_pv_, which represents the height difference between the lowest and the highest points on the scanned area) reached ~101 nm. Sample M1 also exhibited large pores. Between these pores, some areas exhibited hills/valleys morphologies, as exemplified in Figure 2b (with an RMS roughness of 5.8 nm and a peak-to-valley parameter of ~50 nm). Random particles and elongated ditches were also visible on the surface of the M1-L5 sample. Sample M1-L5 appeared to be more compact at lower scales in this series, as exemplified in (Figure 2c), with an RMS roughness of 7.7 nm (given by the presence of protuberance particles/crests) and a peak-to-valley parameter of 46 nm. Sample M1-L10 (Figure 2d) was characterized by a large number of nanometric pores (20–50 nm in diametrer) and randomly distributed larger pores/cavities, as that shown along the horizontal red line which was ~120 nm in diameter. The M1-L10 sample was characterized by an RMS roughness of 5.8 nm and a peak-to-valley parameter of ~53 nm. Sample M2-L5 (Figure 2e) expressed the most uniform morphology, with pores and nearby particles of ~100–150 nm in diameter and it was characterized by an RMS roughness of 16.7 nm and a peak-to-valley value of ~87 nm. Sample M3-L5 (Figure 2f) was the most porous in the series, with rich structures of pores starting from smaller and circular ones (~100 nm in diameter) up to very large and elongated pores of 0.6–1.5 µm in diameter. The image presented in Figure 2f had an RMS roughness of 19.9 nm and an R_pv_ parameter of ~95 nm.

The estimated MWCO value was 1000 Da for M0 and all M1 variants, 600 Da for M2-L5, and 400 Da for M3-L5 membranes, based on the experimental data.

The hydrophilicity of the prepared hybrid nanofiltration membrane had a significant influence on the flux and the separation processes. The contact angle was used as a hydrophilicity/hydrophobicity indicator for membranes. A decrease in the contact angle values was observed when the M0 membrane (PPEES-PVP) (68.4° ± 1.8) (Figure 3) and the other hybrid membranes were analyzed, indicating higher membrane hydrophilicity. This effect can be explained both by the presence of hydrophilic polymers PAN and PVP and SBA-15 silica and by the influence of the laser on the polymer solution. An increase in hydrophilicity could also be observed for a higher concentration of PAN in the casting solution. It was observed that the most efficient action time of the laser was 5 min, for which the M1-L5 membrane had the lowest value of the contact angle.

There was also a close correlation between roughness and the degree of hydrophilicity of the membranes. Thus, for hydrophilic membranes (contact angle < 90°), higher roughness resulted in increased polar interaction with water droplets and a decrease in the contact angle with water.

### 3.2. Antifouling Performance of Membranes

The antifouling properties of the prepared hybrid nanofiltration membranes were investigated by calculating the following indicators: FRR, Rt, Rr, and Rir, presented in Figure 4.

The M1-L5, M2-L5, and M3-L5 exhibited high FRR values, demonstrating antifouling capability and reusability. An increase in Rr value for all hybrid nanofiltration membranes compared to the M0 membrane was also observed, which directly correlates with their hydrophilic properties. The results confirm that the obtained hybrid membranes, which are more hydrophilic, as indicated by the contact angle values, can prevent the attachment of BSA molecules to the membrane surface. These can be removed by back-washing, as reported in other studies [41,42].

### 3.3. Hybrid Nanofiltration Membrane Performance

The performance of the PPEES-PVP and hybrid nanofiltration membranes compared with a commercial nanofiltration membrane was also investigated using the water flux, permeate flux, total polyphenols, and flavonoid rejections (Table 2).

The concentration of phytochemicals from agri-food by-products has attracted special attention in recent years, as natural phytochemicals can be used in pharmaceuticals and cosmetics and also as a low-cost food resource [43]. Apple pomace is an important source of dietary fibers polyphenols, flavonoids, and carotenoids and exhibits antioxidant properties [44]. Recent research has shown the effectiveness of nanofiltration for the concentration of polyphenols from fruit extracts [45,46,47]. Therefore, the aim of the present study was to find new types of membranes that would ensure both high concentration of polyphenols and flavones from apple pomace extract and higher permeate flows.

Our results show that all prepared membranes, both hybrid and PPEES-PVP, have a higher flux for apple pomace extract compared to the commercial reference membrane. The addition of PAN and SBA-15 in the composition of the membranes leads to an increase in their hydrophilicity and permeation, similar to what has been reported in other studies [15,48]. The highest fluxes were obtained for the M1-L10 membrane, but it also had much lower retention of polyphenols and flavones compared to other membranes. This can be explained by the changes in the membrane structure after 10 min of laser treatment where microvoids were detected.

With the increase in PPEES concentration from 13 to 17 wt. %, the separation efficiency for total polyphenols and flavonoids significantly increased. Data from Table 2 showed that hybrid nanofiltration membranes M2-L5 and M3-L5 present a higher rejection coefficient for total polyphenols (78.6 ± 0.7% and 97.8 ± 0.9%, respectively) and flavonoids (28.7 ± 0.2% and 50.3 ± 0.4%, respectively) and are far superior to commercial membrane or M0 membrane. Low solute permeability coefficient values (B) indicate a limitation in the amount of solutes lost in the permeate. The lowest value for the B coefficient was obtained for the M3-L5 hybrid nanofiltration membrane, which correlates with the high values obtained in the retention of polyphenols and flavonoids.

Similar results were obtained by Uyttebroek and collaborators [43] using a commercial hydrophilic NF membrane with MWCO of 150–300 Da.

The separation performance of the prepared nanofiltration membrane for the selected polyphenols was further studied using HPLC (Table 3).

The results obtained using HPLC correlated with those presented in Table 2 regarding the efficiency of the M3-L5 hybrid nanofiltration membrane in concentrating phenolic acids and flavonoids from apple pomace extract.

Figure 5 shows the MgSO_4_ rejection for the prepared hybrid membranes.

Rejections of MgSO_4_ between 15.1% and 56.2% were obtained, the highest value having membrane M3-L5, in accordance with the results observed for nanofiltration polyethersulfone membrane [49].

## 4. Conclusions

In this study, five hybrid nanofiltration membranes based on PPEES, PAN, PVP and SBA-15 mesoporous silica were prepared using phase inversion method. Compared with the PPEES-PVP membrane, the introduction of PAN and SBA-15 into the polymeric casting solution affects the structure and separation performance of the hybrid nanofiltration membranes. Increasing the content of PAN in the membrane composition significantly improves the hydrophilicity of the membrane surface, and the water contact angle of the hybrid membrane decreased from 68.4 ± 1.8° for the M0 membrane to 40.5 ± 0.9° for the membrane based on PPEES/PAN/PVP/SBA-15: 13/7/1/1 exposed to a laser for 5 min (M1-L5). Laser treatment of the polymer solution also had little effect on increasing the hydrophilicity of the membrane, but with a longer treatment time (10 min) it has destructive effects. The antifouling indicators were calculated, and it was found that the introduction of PAN and SBA-15 into hybrid nanofiltration membranes improved their antifouling performance. Evaluation of the concentration of polyphenols and flavonoids in apple pomace extracts showed that the hybrid membrane based on PPEES/PAN/PVP/SBA-15:17/3/1/1 irradiated with a laser for 5 min (M3-L5) had 1.42 times higher rejection of phenolic acids and 2.35 times higher rejection of flavonoid compounds compared to the M0 control membrane. The hybrid nanofiltration membranes also exhibited higher permeate fluxes than the M0 control membrane. All these results demonstrate that hybrid nanofiltration membranes, especially membranes based on PPEES/PAN/PVP/SBA-15:17/3/1/1 laser irradiated for 5 min, are an excellent choice for the concentration of polyphenols and flavones from apple pomace extracts.

## Figures and Tables

**Figure 1 membranes-12-00689-f001:**
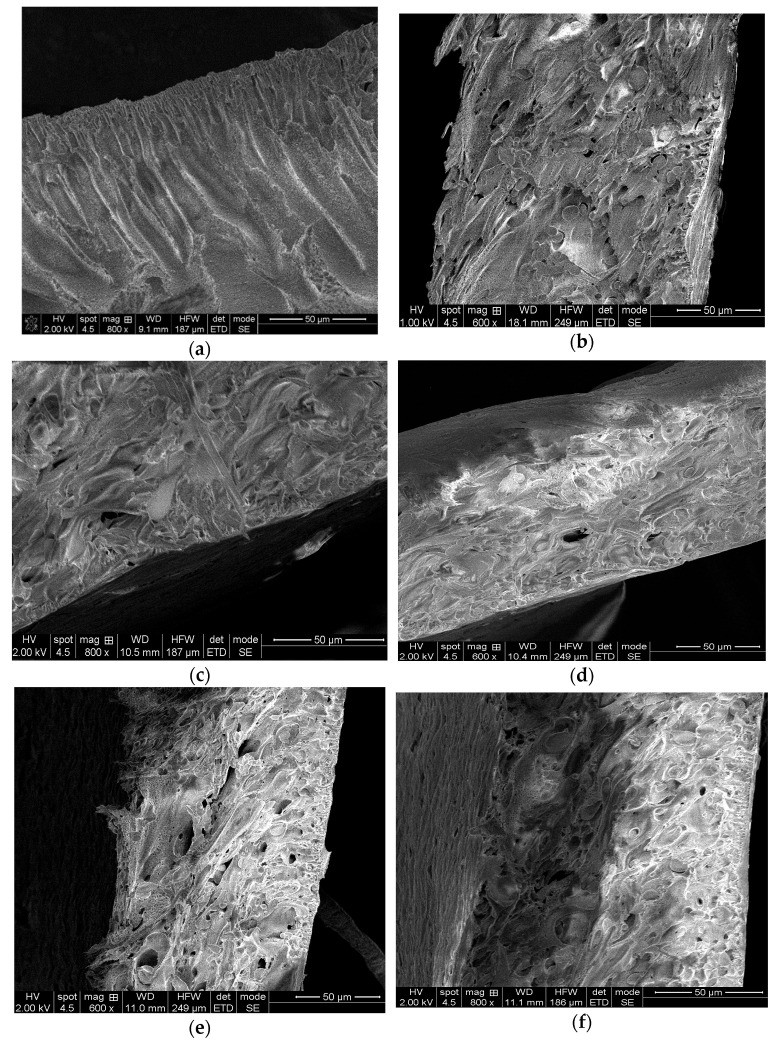
SEM cross-sectional images for the obtained membranes, where: (**a**) M0 control membrane; (**b**) M1 membrane; (**c**) M1-L2-M1 membrane exposed to laser for 2 min; (**d**) M1-L5-M1 membrane exposed to laser for 5 min; (**e**) M1-L10-M1 membrane exposed to laser for 10 min; (**f**) M2-L5-M2 membrane exposed to laser for 5 min; and (**g**) M3-L5-M3 membrane exposed to laser for 5 min.

**Figure 2 membranes-12-00689-f002:**
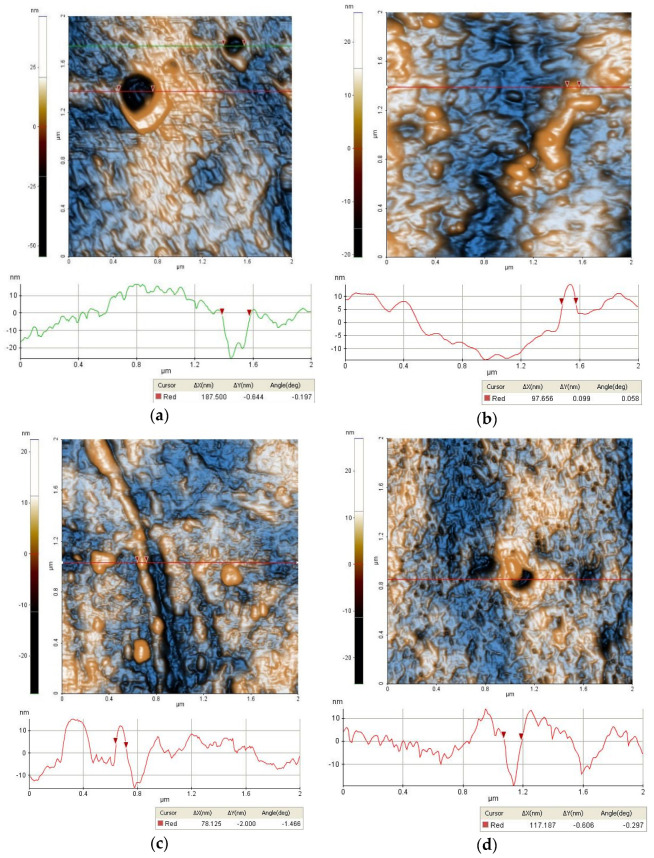
Bidimensional 2D AFM images (presented in enhanced-contrast view mode) together with characteristic line scans recorded from each image at the position indicated in each figure of (**a**) M0 membrane; (**b**) M1 membrane; (**c**) M1-L5 membrane; (**d**) M1-L10 membrane; (**e**) M2-L5 membrane; and (**f**) M3-L5 membrane.

**Figure 3 membranes-12-00689-f003:**
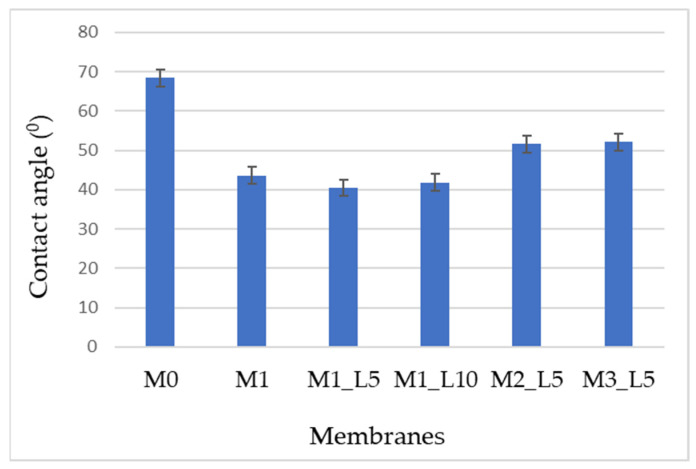
Contact angles for the prepared membrane. All results are significantly different from each other (*p* < 0.05).

**Figure 4 membranes-12-00689-f004:**
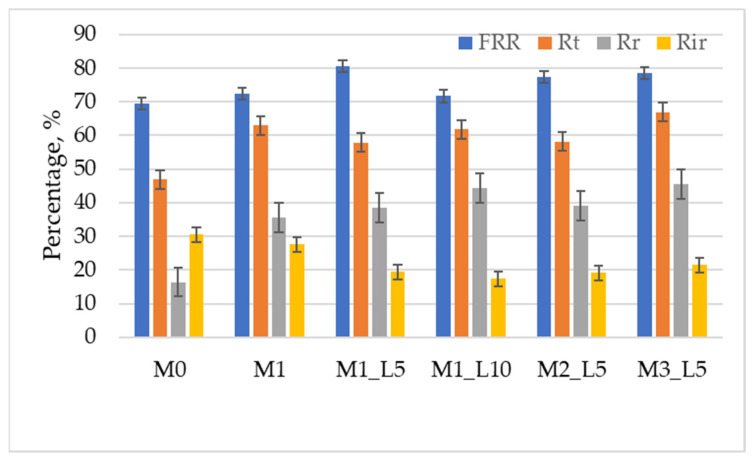
Fouling parameters of membranes with BSA as pollutant.

**Figure 5 membranes-12-00689-f005:**
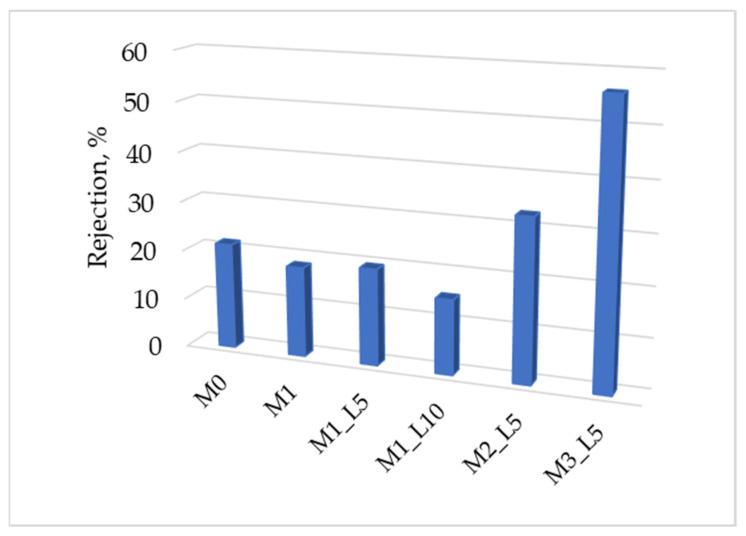
Salt rejection.

**Table 1 membranes-12-00689-t001:** Membrane composition.

Membrane Code	Composition (wt. %)			
	PPEES	PAN	PVP	SBA-15
M0	20	0	1	0
M1	13	7	1	1
M1-L2 ^1^	13	7	1	1
M1-L5 ^2^	13	7	1	1
M1-L10 ^3^	13	7	1	1
M2-L5 ^1^	15	5	1	1
M3-L5 ^1^	17	3	1	1

^1^ M_n_-L2–membrane exposed to laser for 2 min; ^2^ M_n_-L5–membrane exposed to laser for 5 min; ^3^ M_n_-L10–membrane exposed to laser for 10 min.

**Table 2 membranes-12-00689-t002:** Permeation performance and rejection for total polyphenols and flavonoids from apple pomace extract for the prepared membranes.

Membrane Type	Permeate Flux ^a^ (Lm^−2^h^−1^)	Total Polyphenols		Flavonoids	
		Rejection (%)	Solute Permeability, B	Rejection (%)	Solute Permeability, B
Millipore	10.7 ± 0.1	57.1 ± 0.4	7.21 ± 0.05	18.2 ± 0.1	43.09 ± 0.3
M0	11.1 ± 0.1	68.8 ± 0.3	4.48 ± 0.03	21.4 ± 0.3	36.27 ± 0.2
M1	11.7 ± 0.1	65.9 ± 0.5	4.93 ± 0.04	15.6 ± 0.1	56.16 ± 0.4
M1-L5	40.8 ± 0.3	74.2 ± 0.6	4.89 ± 0.02	18.5 ± 0.2	68.99 ± 0.6
M1-L10	106.1 ± 6.2	60.7 ± 0.4	23.76 ± 0.1	10.8 ± 0.1	267.95 ± 1.6
M2-L5	25.2 ± 0.1	78.6 ± 0.7	4.25 ± 0.03	28.7 ± 0.2	48.70 ± 0.4
M3-L5	23.7 ± 0.2	97.8 ± 0.9	0.41 ± 0.01	50.3 ± 0.4	18.33 ± 0.2

^a^ Obtained through filtration of pure water and extracted at 8 bar.

**Table 3 membranes-12-00689-t003:** HPLC-MS analysis for apple pomace retentate fractions.

Compound	Feed, μg/mL	Retentate M0, μg/mL	Retentate M1, μg/mL	Retentate M1-L5, μg/mL	Retentate M1-L10, μg/mL	Retentate M2-L5, μg/mL	Retentate M3-L5, μg/mL
Ellagic acid	3.55	3.66	3.76	3.78	3.64	3.71	4.97
Chlorogenic acid	1.21	1.61	1.47	1.49	1.92	1.39	1.68
(+) Catechin	1.76	1.99	2.11	2.23	0.66	2.67	3.66
Rutin	0.58	0.59	0.56	0.65	0.50	1.10	1.27
Quercetin-3-β-D-qlucoside	1.69	2.15	1.81	2.61	1.67	2.77	3.26
Quercitrin	12.8	13.85	13.48	14.78	14.07	12.7	20.57
Quercetol	0.73	1.00	0.98	0.77	0.75	0.95	1.07
Phloridzin	2.65	2.69	3.13	3.08	2.90	2.84	4.14

## Data Availability

Not applicable.

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
