# Peer review of "New Hybrid Nanofiltration Membranes with Enhanced Flux and Separation Performances Based on Polyphenylene Ether-Ether-Sulfone/Polyacrylonitrile/SBA-15"

_membranes, 2022, doi:10.3390/membranes12070689_

Round 1

Reviewer 1 Report

The manuscript seems very minimal and very little attention has been paid to the english grammar or presentation of data. The manuscript in the current form cannot be published and must be revised thoroughly for reconsideration. 

1. Abstract and the rest of the manuscript. Please revise it thoroughly for grammar and other mistakes. it is very difficult to read and understand 

2. Line 15. please correct this statement to "The addition of.."

3. Line 22. should be 78.6 +- 0.7% and 97.8 +- 0.9%, respectively).

4. Line 22, should be 28.7 +- 0.2% and 50.3 +- 0.4 %, respectively).

5. Line 23, "Commercial membrane..."

6, Keywords. why are there semicolons and commas? Please be consistent

7. Line 30 - 32, please add references to each application mentioned.

8. Line 34 - 36, Sudden transition from agro-industrial by-products to polyphenolic compounds may be confusing for readers Please specify why polyphenolic compounds are important. In general, the problem to be addressed or the basis for research are not clearly explained. 

9. Lines 40-42, too many commas and a long sentence. Please review English language and rewrite concise clear sentences. 

10. Lines 43 - 46, Please cite references for each example. The references should be from the last 3-4 years and preferably from the membranes journal.

11. Line 54, change "resistance" to "resistant"

12. Line 58, remove  "respectively"

13. Line 70, literature examples [13-18] or  need to be specifically mentioned from each reference e.g. please cite the reference for inorganic coating or organic coating separately and individually instead of lumping all references after general statements.

14. Line 72, references [18-21]. Please cite specific examples of how the Silica was used in different membranes and how it helped improve the membrane properties. Each example should accompany their respective article reference. instead of 18-21 references together, please summarize how the Sio2 helped , at least from one or two references .

15. lines 80-82. There is no explanation of the purpose of using laser on the polymer solution. Please elaborate why the laser was used. 

16. Lines 104 and 107, please correct "mixt"

17. Table 1. There is no control experiment. Controls of M0 with Laser

M2-L2, M2-L10, M3-L2 and M3-L10 are missing. 

18. In general, how were the pore size and pore size distribution measured and is the data available. How did the authors conclude that the membranes are nanofiltration membranes without the pore size data?

19. Figure 1. M0 Why is M0 shown at 500 um scale while others are at 50 um scale? Also please replace "um" with proper "micron" symbol.

20 Figure 1 needs to be captioned properly. Please describe what each image corresponds to. 

21. Line 217. 5 x5 um2 samples are not shown. Please clarify why or just delete the reference.

22. Line 217. "Since the of the pores...." Please correct the sentence.

Reviewer 2 Report

The paper presents a number of different membranes that were prepared by treating the polymers with a laser before the membranes are formed.

The language in the paper is generally clear. However, there is quite a number of minor grammatical errors. The paper would benefit from a careful revision of grammar. Furthermore, it appears that in the abstract the font size is not consistent.

·       It appears that the laser treatment is one of the main differences between the membranes. However, the fact that different laser treatments are compared is not mentioned in abstract or title.

·       The abstract contains abbreviations M2_L5 and M3_L5. It would be better to replace these with something descriptive, so that the abstract can be understood without reading the paper.

·       L108 Laser treatment; as it is described, the method cannot be reproduced. Additional details should be added, for example:  what does “some sample” mean? Size of container, how was it mixed etc etc. It would also be beneficial to mention what the purpose is of the laser treatment.

·       Membrane fouling evaluation L146

o   Method needs more details. Such as: what apparatus/equipment was used? What was the pressure? Was it dead-end (with or without stirrer, rpm?) or was it crossflow (what velocity)?

o   Normalized flux and normalized flux recovery should not be used to compare different membranes.  (Suggested is to read (no need to cite): Blankert, B.; Van der Bruggen, B.; Childress, A.E.; Ghaffour, N.; Vrouwenvelder, J.S. Potential Pitfalls in Membrane Fouling Evaluation: Merits of Data Representation as Resistance Instead of Flux Decline in Membrane Filtration. Membranes 2021, 11, 460. https://doi.org/10.3390/membranes11070460), and consider representing the fouling data as resistance and filtered volume.

·       L281: what is the difference between water flux and permeate flux? Table 2 what is the meaning of extract flux?

·       Rejection depends on membrane properties, but also strongly depends on filtration flux, and somewhat on crossflow velocity. Since the fluxes appear to vary quite a bit, it seems inappropriate to compare rejection in this way. It would be better to (also) calculate a solute permeability (“B”).  

·       The abstract mentions that the developed membranes are better than a commercial one. However, it appears that the commercial membrane did not participate in most of the tests. It is also not clear if the commercial membrane can be expected to be particularly suitable for the application under consideration.

·       Conclusions: since rejection is typically expressed as %, it is confusing and potentially misleading to say that X has 71% more rejection than Y.

·       Conclusions: it is suggested to replace abbreviations M1_L5 etc. by something descriptive, so that conclusions can be read stand-alone.

Round 2

Reviewer 1 Report

Even though authors have edited the previous version of the manuscript as per the reviewers comments, there are still lot of grammatical errors in the whole text. The manuscript needs to be thoroughly reviewed for grammar and typographical errors. They have not changed the nomenclature (ML) to something more descriptive of the changes, as suggested by another reviewer. It is difficult to accept the current version with poor English. I recommend that the English be reviewed before consideration for publication. 

Author Response

We corrected all the grammatical errors in the manuscript and highlighted them in red.

We would like to thank you for your review and for providing suggestions to improve the quality of our paper.

Yours Sincerely,

Prof. Gabriel Lucian RADU

Reviewer 2 Report

Although my comments were not fully addressed, I believe that the paper has sufficiently improved to justify publication in Membranes.

It is suggested to make another pass with spelling/grammar checker; e.g. omogeneous solution (L117)

Author Response

We appreciate the time and effort that you have dedicated to providing your valuable feedback on our manuscript. We would like to thank you for your review and suggestions for improving the quality of our work.
We corrected all the grammatical errors in the manuscript and highlighted them in red.

Sincerely,

Prof.Dr. Radu Gabriel Lucian